# Macropinocytosis: Both a Target and a Tool for Cancer Therapy

**DOI:** 10.3390/biom15070936

**Published:** 2025-06-26

**Authors:** Manhan Zhao, Liming Zhou, Yifei Zhai, Aiqin Sun, Genbao Shao, Qiong Lin

**Affiliations:** School of Medicine, Jiangsu University, 301 Xuefu Road, Zhenjiang 212013, China; 2212313096@stmail.ujs.edu.cn (M.Z.); 2212313098@stmail.ujs.edu.cn (L.Z.); 2212413005@stmail.ujs.edu.cn (Y.Z.); aiqinsun@ujs.edu.cn (A.S.); gbshao07@ujs.edu.cn (G.S.)

**Keywords:** macropinocytosis, drug delivery, cancer therapy, therapeutic targeting

## Abstract

Macropinocytosis is a non-selective, clathrin-independent endocytic process that facilitates bulk internalization of extracellular fluid and its dissolved components (including proteins, lipids, and nucleotides) through plasma membrane remodeling and the subsequent formation of macropinosomes. This evolutionarily conserved cellular process plays important roles in nutrient supply, immune response, and metabolism. Particularly, cancer cells exploit activated macropinocytosis to obtain nutrients for supporting proliferation and survival under nutritional stress. Thus, macropinocytosis emerges as an important target for cancer therapy. Furthermore, as activated macropinocytosis constitutively uptakes extracellular fluids into cancer cells, it has been utilized for delivering anti-tumor drugs in cancer therapy. In this review, we systematically addressed progress in cancer therapeutic strategies in both targeting macropinocytosis and utilizing macropinocytosis as an anti-cancer drug delivering tool, including therapeutic applications with macropinocytosis inhibitors; metabolic modulators; methuosis (the macropinocytosis-associated cell death) inducers; and macropinocytosis-mediated anti-cancer drug delivery strategies such as nanoparticles, viral vectors, extracellular vesicles, and targeted conjugates. We conclude that developing targeted macropinocytosis anti-cancer drugs and exploring macropinocytosis-dependent anti-cancer drug delivery systems open new avenues for cancer therapy.

## 1. Overview of Macropinocytosis

Macropinocytosis is an evolutionarily conserved cellular process. Macropinocytosis, a non-coating plasma membrane endocytosis, differs from the clathrin-dependent endocytic pathway. As shown in Figure 1, macropinocytosis contains four stages: (1) initiation and activation; (2) formation of macropinosomes; (3) maturation; and (4) degradation and recycling of macropinosomes [1]. Macropinocytosis is initiated and activated by nutritional stress signals, extracellular stimuli such as growth factors, or chemokines [2,3]. These signals primarily activate downstream signaling pathways involving small GTPases RAS and RAC1 and inositol phospholipid kinases PI3K and PI4K. These pathways modulate the actin cytoskeleton, which drives plasma membrane remodeling [4,5] and leads to the formation of cup-shaped membrane ruffles [6,7]. Subsequently, these ruffles extend distally and wrap the extracellular fluids containing proteins, lipids, and nucleotides by either fusing with each other or folding back onto the plasma membrane, ultimately forming macropinosomes [2]. During maturation, early macropinosomes shrink in size regulated by ion channels to become late macropinosomes [1] and are eventually transported to lysosomes for degradation or recycled back to the plasma membrane. The degraded biomolecules provide nutrients for cancer cell growth. Notably, macropinocytosis in cancer cells with oncogenic *RAS* mutations is constitutively activated and utilized for supporting tumor growth [8]. Thus, oncogenic *RAS* mutation-bearing tumors are the most suitable for targeted therapies harnessing macropinocytosis.

There are multiple excellent review articles published about the molecular mechanism underlying macropinocytosis and the signaling pathways regulating macropinocytosis [2,9,10]. Thus, here we focus on a review of the current progress in macropinoctosis-related cancer therapeutic strategies, including both targeting macropinocytosis to inhibit tumor growth and utilizing macropinocytosis as an anti-cancer drug delivery tool. The content of these two aspects includes therapeutic applications with macropinocytic inhibitors; methuosis-inducing agents; and the macropinocytosis-mediated anti-cancer drug delivery strategies such as nanoparticle-conjugated anti-cancer drugs, virus-loaded anti-cancer protein cDNAs, and exo-vesicle (liposome)-loaded anti-cancer drugs. In conclusion, macropinocytosis offers new cancer therapeutic strategies and techniques based on the cellular function of macropinocytosis in providing nutrients for tumor growth and delivering extracellular fluids into cancer cells.

## 2. Roles of Macropinocytosis in Cancers

### 2.1. Providing Nutrients for Cancer Cell Growth

(a)Uptake of proteins and ATP: It has been observed that the majority of biomolecules in macropinosomes are proteins [11]. These proteins are digested into free amino acids in lysosomes that are used to synthesize new proteins or catabolized to generate ATP for energy [6]. The main protein internalized by macropinocytosis is extracellular albumin, the most abundant protein in human plasma (~50%) [12]. In addition, given that albumin serves as a carrier protein, albumin-bound molecules such as fatty acids (FAs) and cholesterol, which are essential components for maintaining the structural integrity of the cell membrane, are also internalized by macropinocytosis [12]. Apparently, macropinocytosis enables tumor cells to sustain their metabolic and biosynthetic needs and promotes tumor cell growth under nutritional stress conditions [11,13].

It has been observed that extracellular ATP is extensively internalized by macropinocytosis in cancer cells [14,15]. This internalized ATP provides an important source of energy for tumor proliferation, survival, and resistance to targeted therapies [15]. Consequently, suppressing macropinocytosis effectively diminishes extracellular ATP uptake in cancer cells, resulting in depleting the cells’ energy reservoirs and impeding tumor progression in vivo. Emerging evidence further reveals an important role of ATP in driving epithelial–mesenchymal transition (EMT) in a macropinocytosis-dependent manner [15].

(b)Uptake of necrotic cell debris: In addition to extracellular proteins and ATP, cancer cells assimilate other extracellular bioactive macromolecules through macropinocytosis. Tumor cell necrosis occurs under hypoxic or nutrient-deprived conditions and generates abundant cell debris containing proteins, fatty acids (FAs), nucleotides, and triacylglycerols (TAGs) [16,17]. These necrotic fragments are captured by neighbor tumor cells via macropinocytosis and processed by lysosomes to produce nutrients for fueling tumor cell growth. In addition, the uptake of necrotic cells by macropinocytosis may also contribute to resistance against antimetabolite chemotherapeutics [18].

### 2.2. Counteracting Oxidative Stress

Malignant tumor cells frequently exhibit disrupted redox homeostasis due to their heightened metabolic activity combined with microenvironmental hypoxia, leading to the sustained accumulation of reactive oxygen species (ROS) [19,20]. To counteract oxidative stress, cancer cells have evolved multiple defensive mechanisms, including macropinocytosis [2]. Macropinocytosis alleviates ROS-induced cellular damage through the internalization of exogenous anti-oxidants (e.g., glutathione precursors) and redox-regulating metabolites (e.g., cysteine) for endogenous anti-oxidant synthesis [21].

It has been shown that pharmacological blockade of autophagy in pancreatic ductal adenocarcinoma (PDAC) undergoes metabolic reprogramming by upregulating macropinocytosis activity, and the nutrient acquisition switches from autophagy to macropinocytosis [21]. This adaptive metabolic reprogramming is mediated by the nuclear factor erythroid 2-related factor 2 (NRF2) anti-oxidation pathway that upregulates macropinocytotic genes to activate the salvage pathway. NRF2 is a known transcriptional regulator in response to oxidative stress, governing the expression of genes involved in anti-oxidation and redox homeostasis [2,22]. The activation of macropinocytosis by NRF2 upon the inhibition of autophagy may represent a new signaling pathway that counteracts cellular nutritional and/or oxidative stress.

### 2.3. Promoting Immune Escape of Cancers

Macropinocytosis plays a key role in cancer immune escape by regulating the tumor microenvironment (TME) and immune cell function [23,24]. First, cancer cells utilize macropinocytosis to internalize extracellular proteins, which are lysosomally degraded to generate immunosuppressive metabolites, particularly lactate and adenosine, via enhanced glycolytic flux and purine salvage pathways [25]. The accumulation of lactate in the tumor microenvironment promotes immune escape of tumor cells by reducing immune cell activity and inhibiting the production of inflammatory mediators [26]. Secondly, macropinocytosis may affect the immune escape process by regulating the expression of immune checkpoint molecules (e.g., PD-L1). It was observed that macropinocytosis in cancer cells activates the mTOR signaling pathway, which promotes the expression of PD-L1 and inhibits T cell activation and function, thereby leading to immune escape of cancer cells [27,28,29]. In addition, macropinocytosis compromises the antigen-presenting function of tumor-associated macrophages (TAMs) and dendritic cells (DCs), reduces their immune surveillance capacity against cancer cells and further promotes immune escape through the internalization of the antigenic molecules [30,31]. Therefore, targeting macropinocytosis has become a new strategy to enhance the efficacy of immunotherapy. For example, the combination of macropinocytosis inhibitors with immune checkpoint inhibitors (e.g., anti-PD-1/PD-L1 antibodies) can produce synergistic anti-tumor effects [32,33]. This strategy provides a new direction for cancer therapy by simultaneously blocking the macropinocytosis-mediated immune escape pathway and enhancing the anti-tumor activity of immune cells.

### 2.4. Induction of Cancer Cell Death (Methuosis) upon Excessive Macropinocytosis

The term “methuosis” was first defined by Overmeyer et al. in glioblastoma cells with oncogenic *RAS* hyperactivation that stimulates excessive macropinocytosis and leads to cancer cell death. Thus, methuosis is a form of cell death that is induced by the over-activation of macropinocytosis and characterized by the formation of large vacuoles [34]. The large vacuoles [34], which are derived from the fusion of macropinosomes upon the excessive activation of macropinocytosis, are unable to be transported to lysosomes. The lack of lysosomal degradation results in the expansion and accumulation of the large vacuoles in cells [35]. Consequently, the accumulation of large vacuoles segregates essential nutrients and organelles, leading to metabolic stress and energy depletion. Furthermore, the expansion of large vacuoles disrupts cellular structure and function and ultimately leads to cell death [36,37,38]. Methuosis differs from apoptosis as it does not have significant nuclear condensation during the process and is not dependent on the activation of caspase family members [39]. In addition, methuosis does not exhibit the increase in membrane permeability seen in necrosis at its early stage [34]. Methuosis has been observed in cells from multiple types of cancers [40,41,42], offering a novel perspective on cancer therapeutic interventions. However, the precise molecular mechanisms underlying methuosis and the regulatory pathways of methuosis are yet to be fully elucidated.

## 3. The Role of Macropinocytosis in Cancer Therapy

Macropinocytosis is an important cellular process for cancer cell growth and survival under nutritional stress. Targeting macropinocytosis emerges as a promising therapeutic strategy for cancers, especially for cancers with oncogenic *RAS* mutants that have active macropinocytosis. On the other hand, macropinocytosis, as a process that uptakes extracellular fluids into cells, can be used for anti-cancer drug delivery in cancer therapy. Thus, macropinocytosis has dual roles in cancer therapeutic applications: both as a therapeutic target and as a therapeutic tool.

### 3.1. Targeting Macropinocytosis for Cancer Therapy

#### 3.1.1. Application of Macropinocytosis Inhibitors for Cancer Therapy

The development of inhibitors of macropinocytosis as targeted cancer therapeutic drugs may effectively impede the macropinocytosis-dependent oncogenic pathways and disrupt the metabolic activity of cancer cells. As shown in Table 1, several inhibitors of macropinocytosis have been developed. Most of the inhibitors attack the early stage (initiation and activation stage) of macropinocytosis.

The small GTPase RAC1, a pivotal regulator in the initiation of macropinocytosis, has emerged as a significant drug target. Research findings have demonstrated that the inhibition of RAC1 activity can effectively impede the process of macropinocytosis. Currently, two compounds, NSC23766 and EHT1864, have been extensively utilized as RAC1 inhibitors in preclinical studies, exhibiting high specificity in inhibiting RAC1 activity and demonstrating therapeutic benefits in *RAS*-mutant cancers, including pancreatic cancer (PDAC) and non-small-cell lung cancer (NSCLC) [43,44]. Among them, NSC23766 specifically inhibits the interaction between RAC1 and guanine nucleotide exchange factor (GEF), preventing the conversion of RAC1-GDP to RAC1-GTP. In contrast, EHT1864 binds RAC1 isoforms (RAC1b, RAC2, RAC3) with a high affinity, inducing nucleotide dissociation [44,45]. Recent studies have found that the N,N′-disubstituted guanidino analog 1D-242 significantly inhibits RAC1-mediated TNFα-induced NF-κB nuclear translocation during non-small-cell lung cancer (NSCLC) cell proliferation and migration [46]. Furthermore, the E3 ubiquitin ligase MG53 has been shown to impede RAC1 activity through a direct targeting approach, leading to ubiquitination modification at the Lys5 site. This, in turn, has been observed to hinder the progression of hepatocellular carcinoma (HCC) [47].

In addition to the direct targeting of RAC1, the PI3K/AKT/mTOR signaling pathway, as an important regulatory pathway of macropinocytosis, has also become an important target for drug development. A notable example is the natural product Wortmannin, derived from *Penicillium*, which has been shown to inhibit cell growth and induce apoptosis in cancer cells by targeting the PI3K/AKT pathway [48]. LY294002, a synthetic compound designed based on the flavonoid quercetin, functions as a PI3K inhibitor, blocking PI3K/AKT signaling, inhibiting glycolysis, and interfering with ATP production, thereby inducing apoptosis in cancer cells [49].

Other types of macropinocytosis inhibitors have also demonstrated promising applications. 5-(N-Ethyl-N-isopropyl)-amiloride (EIPA), an Na^+^/H^+^ ion channel inhibitor, has been shown to inhibit the activities of RAC1 and Cdc42 by lowering the submembranous pH, thereby blocking macropinocytosis and actin polymerization [50]. Cytochalasin D, an inhibitor of actin polymerization, effectively inhibited membrane ruffling and macropinocytosis in fibroblasts [9]. These studies provide a significant theoretical foundation and experimental basis for the development of novel targeted drugs for the therapy of macropinocytosis-associated cancers.

**Table 1 biomolecules-15-00936-t001:** Macropinocytosis-targeted drugs or agents for cancer therapy.

Inhibitor Type	Drug	Mechanism	Cancer Type	References
Macropinocytosis inhibitors	NSC23766	Inhibition of Rac1 activity	*Ras*-mutant tumors	[44]
	EHT1864	Inhibition of Rac1 activity	*Ras*-mutant tumors	[45]
	1D-242	Inhibition of Rac1-mediated nuclear translocation	*Ras*-mutant tumors	[46]
	MG53	Ubiquitination modification at Lys5	*Ras*-mutant tumors	[47]
	Wortmannin	Inhibition of PI3K/AKT pathway	*Ras*-mutant tumors	[48]
	EIPA	Lowering the pH value below the plasma membrane	*Ras*-mutant tumors	[49]
	LY294002	Blocking PI3K/AKT signaling	*Ras*-mutant tumors	[50]
	Cytochalasin D	Inhibition of actin polymerization	*Ras*-mutant tumors	[9]
	NaN3	Blocking ATP dependence	*Ras*-mutant tumors	[51]
	Poziotinib	Targeting EGFR	*Kras*-mutant tumors	[52]
	Ivermectin	Inhibiting PAK1	*Kras*-mutant tumors	[52]
	Tyrphostin A9	Targeting PDGFR	*Kras*-mutant tumors	[53]
	LY2090314	Inhibiting GSK-3	*Kras*-mutant tumors	[54]
	Pyrvinium pamoate	Inhibiting Wnt/β-catenin	*Kras*-mutant tumors	[55]
Metabolic regulators	CQ, HCQ	Inhibition of cytoprotective autophagy	Pancreatic cancer, etc.	[56]
	Lys05	Lysosomal deacidification	Glioblastoma	[57]
	Bafilomycin A1	V-ATPase inhibitors	Various cancers	[58]
	Telaglenastat	Inhibition of glutamine conversion	Melanoma	[59]
Methuosis inducers	MIPP, MOMIPP	Inducing plasma membrane shrinkage	Glioblastoma, Breast cancer	[60,61]
	Bacoside A	Induction of macropinocytosis	Glioblastoma	[62]
	Nerve growth factor	Induction of macropinocytosis	Medulloblastoma	[63]
	Trehalose	Forced induction of macropinocytosis	Glioblastoma	[64]
	12A	Induction of macropinocytosis	Breast cancer, etc.	[65]

#### 3.1.2. Macropinocytosis-Associated Metabolic Regulators

Following the process of macropinocytosis, cancer cells can metabolize internalized proteins, carbohydrates, and other nutrients to generate energy and synthesize precursors for cellular biosynthesis, which participate in various metabolic processes within the cell. Targeting these metabolic pathways can suppress macropinocytosis, deplete energy reserves, and ultimately induce cell apoptosis. The potential drugs used to target macropinocytosis-associated metabolic pathways are listed in Table 1. Notably, autophagy and macropinocytosis exhibit cross-regulation in metabolic processes. The inhibition of autophagy may impede the degradation and recycling of internalized substances in macropinocytosis, consequently resulting in the death of cancer cells due to nutrient deprivation. Among autophagy inhibitors, chloroquine (CQ) and hydroxychloroquine (HCQ), which are widely used to alleviate acute and chronic inflammatory diseases, have been shown to inhibit cytoprotective autophagy, thereby enhancing the sensitivity of cancer cells to chemotherapeutic drugs [56]. The disruption of lysosomes, pivotal organelles in cellular autophagy and macropinocytosis, can markedly inhibit the degradation of internalized substances. Dimeric chloroquine (Lys05), a novel lysosomal inhibitor, has been shown to accumulate within and deacidify the lysosomes of both cells and tumors and to prevent the degradation of nutrients in macropinosomes [57]. Bafilomycin A1, a macrolide antibiotic, functions as a specific and potent inhibitor of V-ATPases, thereby preventing lysosomal acidification and, consequently, impeding the degradation of substances, affecting cancer cell proliferation [58]. Furthermore, L-glutamine (Gln) is imperative for cancer cell proliferation, and macropinocytosis is a key metabolic pathway for glutamine acquisition by cancer cells. Consequently, the inhibition of glutamine synthesis may prove effective in impeding the glutamatergic signaling of cancer cells. Telaglenastat (CB-839), a highly potent glutaminase inhibitor (GLSi) with a favorable systemic tolerance, has been observed to significantly reduce the glutamine to α-ketoglutarate (α-KG) conversion in tumor cells while concomitantly enhancing the activity of T cell-mediated immunotherapy in the context of combating melanoma [59]. These studies provide a significant theoretical foundation and experimental validation for the development of metabolic modulation-based strategies for macropinocytosis-targeted therapy.

#### 3.1.3. Methuosis Inducers

The over-activation of macropinocytosis causes the accumulation of macropinosome-derived large vacuoles, leading to methuosis, a form of cell death associated with excessive macropinocytosis. The dysregulation of ion channels and transporter proteins in the vacuolar membrane increases permeability and osmotic imbalance and promotes vacuole expansion [35]. In addition, lysosomal dysfunction prevents the degradation of vesicle contents, leading to further vesicle enlargement and cell rupture. The accumulation of large vacuoles segregates essential nutrients and organelles, leading to metabolic stress and energy depletion. Large vesicles can displace cytoplasmic organelles and disrupt cellular structure and function, ultimately leading to cell death. Therefore, inducing the over-activation of macropinocytosis to promote methuosis becomes a promising cancer therapeutic strategy. Potential methuosis inducers are shown in Table 1. Silva-Pavez et al. found that treatment with the casein kinase 2 (CK2) inhibitor silmitasertib induced colorectal cancer cells to produce large numbers of macropinocytosis-derived vacuoles, ultimately leading to methuosis [66]. Similarly, Indole-Based Chalcones (e.g., MIPP and MOMIPP) are able to rapidly induce plasma membrane contraction and promote the formation of macropinosomes. These newly formed macropinosomes neither participate in membrane recycling nor fuse with lysosomes, leading to the vacuolization of cells and thus exerting anti-cancer effects [60,61]. Recent studies have shown that natural trehalose triggers early *RAS* activation in *NF1*-deficient U373-MG cells, and this signaling can subsequently stimulate macropinocytosis, which causes the cells to become overstimulated and ultimately undergo methuosis, leading to cell death [64]. A pyridine-pyrimidine-indole-carbohydrazide derivative, 12A, was designed and synthesized by Wu et al. as a novel inducer, which has antiproliferative and vacuolization-inducing effects on cancer cells similar to those of MOMIPP. Notably, the vacuoles induced by this compound were macropinosome-derived rather than autophagosome-derived. In the MDA-MB-231 xenograft mouse model, 12A exhibited significant tumor growth inhibition, making it a potential candidate for the development of novel anti-cancer therapeutic strategies [65].

#### 3.1.4. Targeting Macropinocytosis for Drug Resistance

Chemotherapy is one of the primary clinical approaches for cancer management. However, chemo-resistance poses a persistent challenge during chemotherapeutic interventions. During chemotherapy treatment, certain cancer cells upregulate the expression of ATP-binding cassette (ABC) efflux transporters, including P-glycoprotein (P-gp/ABCB1), multidrug resistance protein 2 (MRP2/ABCC2), and breast cancer resistance protein (BCRP/ABCG2) [67]. These membrane transporters actively expel diverse chemotherapeutic compounds through ATP-dependent mechanisms, thus causing resistance to chemo-drug treatment by reducing intracellular drug accumulation and diminishing its cytotoxic efficacy [68,69]. Macropinocytosis uptakes extracellular ATP (eATP) and causes a significant increase in intracellular ATP levels. This mechanism directly potentiates the efflux activity of primary ABC multidrug transporters, accelerating the extracellular transport of chemotherapeutic drugs, which consequently diminishes intracellular drug retention [70]. Furthermore, numerous chemotherapeutic agents, including gemcitabine [71], 5-fluorouracil [72], doxorubicin [73], and platinum drugs, are nucleotide-targeted therapies by either targeting key enzymes in the nucleotide biosynthetic pathway or damaging DNA [74]. Macropinocytosis circumvents these nucleotide-targeted therapies via the uptake of extracellular proteins that produce free amino acids for metabolic precursors in nucleotide biosynthesis [75]. Thus, macropinocytosis plays an important role in chemo-resistance. The pharmacological inhibition of macropinocytosis (e.g., EIPA) blocks the macropinocytosis-mediated adaptive pathways for chemo-resistance and synergizes with chemotherapy by the dual suppression of nutrient salvage and drug efflux. For example, in *Ras*-mutant tumors, the combination of a macropinocytosis inhibitor with a chemotherapeutic drug (e.g., gemcitabine) synergistically suppresses tumor progression while delaying acquired resistance [76].

Both macropinocytosis and autophagy are stress-adaptive pathways. When autophagy inhibitors are used to treat pancreatic ductal adenocarcinoma (PDAC), macropinocytosis is activated via the NRF2 pathway to mitigate the cytotoxic effect caused by the inhibition of autophagy, thereby causing therapeutic resistance. Therefore, treatment with inhibitors of both autophagy and macropinocytosis (e.g., chloroquine) significantly enhances the anti-tumor effect [21].

### 3.2. Utilizing Macropinocytosis for Anti-Cancer Drug Delivery

In the field of cancer therapy, the development of drug delivery systems has become one of the key strategies to improve therapeutic efficacy and minimize side effects. Macropinocytosis is a non-selective endocytotic process by which cells can take up large amounts of extracellular fluid and dissolved substances. Thus, macropinocytosis offers an effective way to deliver conventional chemo drugs into cells. The macropinocytosis-mediated drug delivery systems have two forms: carrier-based formulations and conjugate-mediated targeted delivery systems.

#### 3.2.1. The Macropinocytosis-Mediated Carrier Delivery System

A carrier formulation delivery system represents the delivery of drugs that need to be packaged in individual carriers, such as exosomes, nanoparticle carriers, and viral carriers that carry the drug into the carrier cavity to deliver the drug molecule. Table 2 shows a list of the current macropinocytosis-mediated carrier-based anti-cancer drug delivery methods.

(a)Nanoparticles

Lipid nanoparticles (LNPs) are the most widely used delivery carrier. LNPs consist of phospholipid bilayers with low toxicity and low immunogenicity and are capable of encapsulating both hydrophilic and hydrophobic drugs. Gilleron et al. demonstrated by image-based analysis that LNPs are mainly used for cellular internalization through clathrin-mediated endocytosis and macropinocytosis [94]. Currently, liposome-based drug delivery is mainly focused on two major areas: chemotherapeutic drugs (e.g., adriamycin) and nucleic acid drugs (e.g., siRNA). Doxil, a doxorubicin formulation of polyethylene glycol-coated liposomes has been successfully used in the clinic [77]. In the laboratory research phase, several innovative studies have demonstrated the potential of liposomal delivery systems. For example, Wu et al. constructed the fibroblast activation protein (FAP)-responsive liposome FrLip@R, which efficiently delivered all-trans retinoic acid (ATRA) to pancreatic stellate cells (PSCs) in a pancreatic ductal adenocarcinoma (PDAC) model by macropinocytosis. It successfully returned activated PSCs to a quiescent state, disrupting the tumor stromal barrier, and significantly enhanced intratumoral drug delivery efficiency [78].

Zheng et al. developed fucoidan-encapsulated pH-sensitive liposomes (FU-GEM PSLs) for the targeted delivery of gemcitabine, which showed good results in the treatment of pancreatic cancer [79]. In terms of nucleic acid drug delivery, small interfering RNAs (siRNAs) have emerged as a promising strategy for liposomes. The siRNA molecules can effectively reduce the expression of oncogenes or other disease-causing genes in cancer cells by silencing specific genes by promoting the degradation of target mRNAs. Yang et al. successfully developed a lipid complex that can deliver siRNAs to glioma safely, efficiently, and selectively via the nasal route. This delivery system was preferentially internalized by glioma cells along with cellular debris through macropinocytosis, which significantly increased the accumulation of therapeutic agents in tumor tissues [81]. Yan et al. constructed a cancer cell membrane-fused liposome (CLip) containing an siATG5-loaded calcium phosphate (CaP) core, termed CLip@siATG5. This system can effectively regulate the metabolism of pancreatic cancer cells and significantly improve the delivery efficiency of chemotherapeutic drugs [80].

In addition to LNPs, other types of nanoparticles have also been developed as novel drug delivery platforms (Table 2), especially for targeted delivery applications. Among them, albumin and lipoprotein are important carriers for macropinocytosis-mediated therapies due to their unique biological properties. As an ideal drug carrier, albumin is characterized by biodegradability, good biocompatibility, low immunogenicity, etc., and the presence of multiple drug binding sites on its molecular surface makes it suitable for loading multiple therapeutic drugs. In clinical applications, nanoparticle albumin-conjugated paclitaxel (nab-paclitaxel, nab-PTX) has been successfully used for targeted therapy of *KRAS*-mutant cancers (e.g., pancreatic cancer and breast cancer) [82]. Studies have shown that *KRAS*-mutant cancer cells exhibit sensitivity to a variety of albumin-conjugated drugs, including adriamycin [95,96], anti-EGFR antibodies [83], and human β-defensin-2 [84]. Lipoproteins, as another important carrier, are composed of lipids and apolipoproteins and are mainly involved in lipid transport in the organism. In recent years, lipoproteins with a small particle size (e.g., low-density lipoprotein, LDL, and high-density lipoprotein, HDL) have been widely used in the development of drug delivery systems. Similar to albumin, lipoproteins have excellent biocompatibility, biodegradability, and non-immunogenicity and are capable of being internalized by tumor cells as nutrients through macropinocytosis. However, the clinical application of natural lipoproteins is limited by low yields and complex purification processes. To overcome these limitations, biomimetic lipoprotein systems have been developed. For example, paclitaxel associated with LDL-like nanoparticles has entered clinical trials (NCT04148833) [85]. The use of HDL nanoparticles encapsulated with salinomycin can enhance the drug uptake of cervical cancer stem cells (CSCs) through macropinocytosis and effectively kill tumor cells at lower concentrations [86]. In addition, combining calcium phosphate encapsulating siRNA with apolipoprotein E3-reconstituted high-density lipoprotein can achieve efficient intracellular delivery of siRNA through macropinocytosis [87].

In addition to albumin and lipoprotein nanoparticles, a variety of other types of nanoparticles have shown unique promise for application in cancer therapy through macropinocytosis. These nanoparticles utilize different mechanisms of action and offer diverse strategic options for cancer therapy. Ruthenium nanoparticles are novel anti-cancer nanomaterials that act by disrupting the cellular redox balance and the actin cytoskeleton. This dual mechanism of action induces oxidative stress, leading to cytoskeletal reorganization, which in turn impairs the nutrient uptake capacity of cancer cells and ultimately promotes cancer cell death [97,98]. In the field of gene therapy, Su et al. found that the overexpression of heat shock protein A9 (HSPA9/GRP75) significantly enhanced macropinocytosis in ovarian cancer cells. This property synergized with concentrated Tat peptide–plasmid DNA complex (Tat-pDNA) nanoparticles with added Ca^2+^ to significantly enhance the cytotoxic effect on tumor cells [99]. In addition, miRNA-34a-encapsulated nanoparticles could be efficiently taken up by cancer cells via macropinocytosis, thereby inhibiting tumor growth. Liposome–polycation–peptide nanoparticle (LPP) mRNA vaccines based on poly (β-amino ester) polymer mRNA encapsulated in lipid shells have shown excellent delivery efficiency in the presence of macropinocytosis, especially in dendritic cells [88]. In terms of traditional chemotherapy drug delivery, paclitaxel-loaded poly(lactic-co-glycolic acid) nanoparticles exhibited dual cytotoxicity effects on breast cancer cells by macropinocytosis, which retained the anti-cancer activity of paclitaxel itself and enhanced the targeted delivery effect of the nanoparticles [89].

(b)Virus vectors

Adenovirus, lentivirus, and other viral vectors have been used as the anti-cancer drug carriers in the macropinocytosis-mediated drug delivery [90,91] (Table 2). Viral particles first activate signaling pathways that trigger actin-mediated membrane ruffling, followed by the formation of a macropinosome on the plasma membrane, which in turn leads to the internalization of viral particles into the cell. By exploiting this natural infectious ability and efficient intracellular delivery property, drugs can be delivered inside cancer cells by virus vectors. It was observed that lentiviruses employ macropinocytosis as their primary invasion route. For example, human immunodeficiency virus type 1 (HIV-1) enters macrophages through a CCR5 (C-C chemokine receptor 5)-triggered macropinocytosis-like mechanism, subsequently completing its uncoating within the endolysosomal pathway [100]. Macropinocytosis is one of the known endocytic routes for adeno-associated virus (AAV) that is used for anti-tumor drug delivery [101,102]. In addition, nanoparticle–oncolytic virus–polymer composites can be efficiently taken up by cancer cells through macropinocytosis, enhancing targeting and immune activation against cancer cells [91]. Moon et al. utilized a pH-sensitive and bio-reducible polymer (PPCBA) to complex with an oncolytic adenovirus (Ad), forming Ad-PPCBA. The Ad-PPCBA targets the acidic and hypoxic tumor microenvironment and overcomes coxsackie and adenovirus receptor (CAR)-dependent entry into target cells, showing potential for treating both primary and metastatic tumors [91].

(c)Exosomes

Exosomes, a class of nanoscale extracellular vesicles (40–150 nm) with a lipid bilayer structure, mediate intercellular communication under physiological and pathological conditions. These natural nanocarriers can be taken up by cells through various endocytosis pathways, including macropinocytosis, which significantly improves the internalization efficiency of exosomes and provides new ideas for drug delivery. Kamerkar et al. found that enhanced macropinocytosis facilitates exosome uptake in *Kras*-mutant cancers. Compared to synthetic liposomes, exosomes showed superior performance in delivering RNA interference (RNAi) molecules and inhibiting tumor growth [103]. Deng et al. fused CLT (Celastrol)-loaded PEGylated lipids with DC2.4 cell membranes (M-LIP-CLT) to construct a delivery system targeting *Kras*-mutated pancreatic cancer [92]. Nakase et al. explored the application of exosomes in protein delivery. By encapsulating the ribosome-inactivating protein saporin within exosomes, they successfully inhibited cytoplasmic protein synthesis in target cells, thereby effectively suppressing the growth of tumor cells [93].

#### 3.2.2. Macropinocytosis-Mediated Conjugated Targeted Delivery Systems

Unlike a carrier-based delivery system, a conjugated targeted delivery system employs a linker to covalently conjugate a drug to a molecule that recognizes the specific target protein or intracellular site, thus delivering the drug via macropinocytosis to the target protein or intracellular site. Current conjugated targeted delivery systems includes two delivery methods: PDCs (Peptide–Drug Conjugates) and ADCs (Antibody–Drug Conjugates). Table 3 shows a list of the current macropinocytosis-mediated conjugated targeted delivery systems.

(a)Antibody–Drug Conjugates (ADCs)

The design of antibody–drug conjugates (ADCs) is based on the concept of the differential expression of tumor antigens between tumors and normal tissues, allowing for the specific delivery of chemotherapeutic agents to the target tumors. ADCs are known to enter cancer cells through macropinocytosis-mediated internalization, thereby inhibiting tumor cell growth and proliferation. AGS-16C3F, an ADC that targets ectonucleotide pyrophosphatase/phosphodiesterase 3 (ENPP3), is currently in development for treating metastatic renal cell carcinoma [104,105].

(b)Peptide–Drug Conjugates (PDCs)

Peptide–drug conjugates (PDCs) are new types of drug delivery methods, in which the anti-cancer drug is covalently conjugated to the designed peptide via linkers. The PDC delivery techniques are based on the principle of the interaction of the peptides with the target tumor molecule or the induction of macropinocytosis by the peptides. PDCs can achieve precise drug delivery in cancer cells, improve therapeutic efficacy, and reduce toxicity to normal cells [106]. In the application of cell-penetrating peptides (CPPs), these short peptides that mimic the structure of viral peptides facilitate the transmembrane transport of exogenous molecules by inducing macropinocytosis. Among them, the TAT peptide, a peptide derived from HIV-1, is the most widely studied CPP, and when nanoparticles are modified with TAT for drug delivery, they carry a “bystander cargo” that can also be internalized by the cell through macropinocytosis. This property simplifies the drug formulation process as no additional modification of the drug molecule is required [109]. In addition, arginine-rich CPPs have shown promising applications in macropinocytosis-mediated drug delivery. For example, octa-arginine peptide (R8) has been successfully used for siRNA and antigen delivery, while arginine 12-mer peptide (R12) induces macropinocytosis activity and promotes receptor internalization by targeting CXCR4 [107,108,110].

#### 3.2.3. Current Clinical Application of the Macropinocytosis-Mediated Anti-Cancer Drug Delivery for Cancer Therapy

In current clinical applications, macropinocytosis, a non-selective endocytic pathway, is being widely explored for the delivery of anti-cancer drugs, especially in *KRAS*-mutated cancers (e.g., lung cancer), and shows significant potential. The application of macropinocytosis in lung cancer therapy has focused on the development of nanocarrier systems capable of efficiently entering cancer cells via the macropinocytosis pathway, thereby increasing the concentration of drug in the cancer cells while reducing the toxic effects on normal cells. For example, Abraxane (albumin-bound paclitaxel, PTX) is a typical example of a drug that utilizes the macropinocytosis effect for drug delivery [40,111]. Compared to conventional paclitaxel, Abraxane offers the advantages of higher tumor uptake, fewer side effects, and a shorter administration time. Abraxane is currently approved by the U.S. Food and Drug Administration (FDA) and the European Medicines Agency (EMA) for the treatment of advanced breast cancer, pancreatic cancer, and non-small-cell lung cancer (NSCLC) [112].

In ovarian cancer therapy, a research team developed Tat/pDNA/C16TAB (T-P-C) nanoparticles. This delivery vehicle employs the cationic surfactant C16TAB to condense Tat/pDNA nanocomplexes, forming irregular particles characterized by a small size, positive surface charge, and high pDNA encapsulation efficiency. The mechanism underlying the delivery involves the Arf6 GTPase/Rab signaling axis-dependent activation of the macropinocytosis pathway [113].

Although macropinocytosis has great potential for drug delivery, it still faces several challenges in clinical application. For example, how to activate macropinocytosis to improve drug delivery efficiency and how to avoid drug degradation in the macropinocytosis pathway are critical issues that need to be addressed. In the future, through in-depth research on the mechanism of macropinocytosis and the combination of nanotechnology innovations, it is expected that more-efficient and safer anti-cancer therapeutic solutions will be developed, bringing new hope to lung cancer and other cancer patients.

## 4. Conclusions

Macropinocytosis has great potential in cancer therapy. Macropinocytosis enables cancer cells to take up exogenous nutrients and maintain malignant proliferation; thus, it is an important therapeutic target. As macropinocytosis is capable of endocytosing extracellular fluids into cells and active tumors, particularly tumors with oncogenic *RAS* mutations, it can be used as a tool for the delivery of anti-cancer drugs specifically to tumors. The current cancer therapeutic research on macropinocytosis includes targeting macropinocytosis, inducing methuosis, and delivering anti-cancer drugs by macropinocytosis. Currently, the clinical application of macropinocytosis for cancer therapy is in the development stage and faces many challenges. To fully exploit macropinocytosis in cancer therapy, the following aspects need to be improved or completed. First, the specificity of drug delivery must be improved. Although cancer cells usually have higher macropinocytosis activity, normal cells can also take up carrier-loaded or conjugated drugs via macropinocytosis. Therefore, how to improve the specificity of macropinocytosis-mediated drug delivery is an important direction for future research. Second, novel drug carriers for cancers must be developed to improve the delivery efficiency and biocompatibility. Third, the regulation mechanism underlying micropinocytosis must be further understood. Although it is known that RAS, RAC1, PI3K, and other signaling pathways regulate the activation of macropinocytosis, the regulatory signaling pathways for the late stages of macropinocytosis, including macropinosome maturation, recycling and degradation, remain incomplete. Fourth, the clinical application of macropinocytosis for clinical cancer therapy must be accelerated. Although macropinocytosis has shown significant effects in laboratory studies, its clinical application is still lacking. More clinical trials are needed to validate the safety and efficacy of macropinocytosis-mediated drug delivery systems. Finally, macropinocytosis targeting must be combined with other cancer therapeutic techniques (e.g., chemotherapy, radiotherapy, immunotherapy, etc.), which may be valuable for improving the efficiency of current cancer therapy.

## Figures and Tables

**Figure 1 biomolecules-15-00936-f001:**
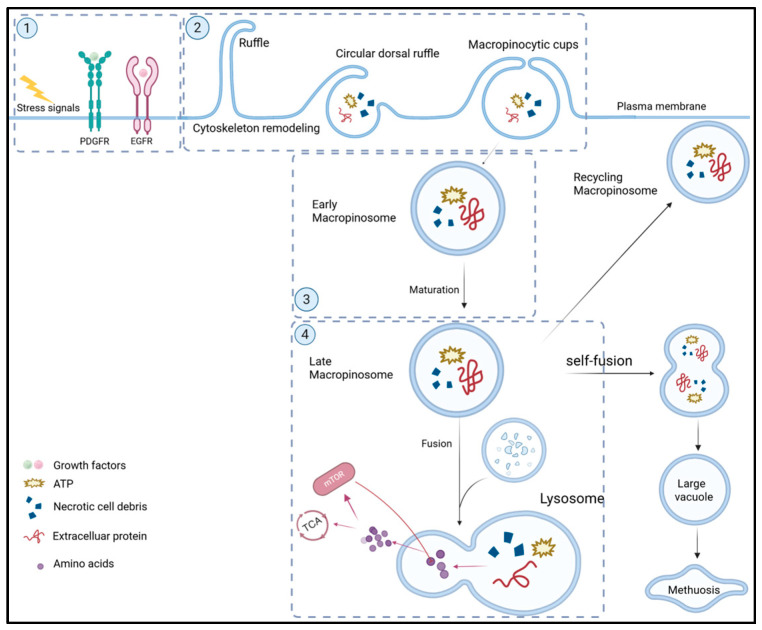
Process of macropinocytosis. The process of macropinocytosis has 4 stages: (1) Initiation and activation: Macropinocytosis is initiated and activated by cellular signals that induce cytoskeletal remodeling to form plasma membrane ruffles. (2) Formation of macropinosomes: The plasma membrane ruffles fold into cup-shaped structures that encapsulate extracellular nutritional fluids and form early macropinosomes. (3) Maturation: The early macropinosomes undergo volume adjustment to become late macropinosomes. (4) Degradation and recycling: The matured macropinosomes either fuse with lysosomes to release nutrients or recycle back to the plasma membrane. Excessive macropinocytosis may generate large vacuoles that lead to non-apoptotic cell death called methuosis.

**Table 2 biomolecules-15-00936-t002:** Macropinocytosis-mediated carrier delivery systems.

Subtype	Carrier	Payloads	Cancer Type	Reference
Nanoparticles	Liposomes	Adriamycin	Kaposi sarcoma	[77]
		All-trans retinoic acid	Pancreatic ductal adenocarcinoma	[78]
		Gemcitabine	Pancreatic cancer	[79]
		siRNA(ATG5)	Pancreatic cancer	[80]
		siRNA	Glioma	[81]
	Albumin	Paclitaxel	Multiple types of cancer cells	[82]
		Adriamycin	Pancreatic cancer	[83]
		Anti-EGFR antibodies	Pancreatic cancer	[83]
		β-Defensin-2	Multiple types of cancer cells	[84]
	Lipoprotein	Paclitaxel	Multiple types of cancer cells	[85]
		Salinomycin	Cervical cancer	[86]
		siRNA	Glioblastoma	[87]
	Lipid shell	mRNA	Melanoma	[88]
		Adriamycin	Breast cancer, etc.	[89]
Virus vectors	Lentivirus	siRNA	Leukemia	[90]
	Oncolytic viruses	PPCBA	Primary tumor and metastatic tumors	[91]
Exosomes	Exosomes	Celastrol	*Kras*-mutant tumors	[92]
	Exosomes	Saporin	*Ras*-mutant tumors	[93]

**Table 3 biomolecules-15-00936-t003:** Macropinocytosis-mediated conjugated targeted delivery systems.

Subtype	Carrier	Payloads	Cancer Type	References
Antibody–Drug Conjugates (ADCs)	Antibody	Chemotherapy drugs	Multiple types of cancer cells	[104,105]
Peptide–Drug Conjugates (PDCs)	Elastin-like peptide	miRNA-34a	Glioma	[106]
	R8	NAP	Multiple types of cancer cells	[107,108]
	R12	NAP	Multiple types of cancer cells	[107,108]

## Data Availability

Not applicable.

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
