# Peer review of "Macropinocytosis: Both a Target and a Tool for Cancer Therapy"

_biomolecules, 2025, doi:10.3390/biom15070936_

Round 1
Reviewer 1 Report
Comments and Suggestions for Authors
This is a nice review on the role of micropinocytosis in cancer and cancer therapy. The article is nice to read. The only major problem is that many sentences stating interesting things have no reference. The authors need to put a reference for each sentence that states some sort of fact.
Other points (listed by line number)
20 ‘methuosis’ is a sufficiently unusual term that you need to explain it here or simplify and explain later
40 ‘Macropinosomes are matured by adjusting the size’ – please explain this
59 ‘This is a figure’ needs to be replaced with something more descriptive
Many sentences need references, examples are those starting on lines 77 and 80, 98, 114
170 ‘This is a table’ needs to be replaced with something more descriptive
Same for 292
265 ‘adenovirus-mediated αvβ5 integrins’ - by ‘mediated’ do you mean encoded?
275 please spell out CAR
348 what is Trop-2?
Reviewer 2 Report
Comments and Suggestions for Authors
The paper by Zhao et al. is focused on macropinocytosis, a non selective internalisation process normally employed by cells for uptake of nutrients, but also involved in cancer cell defense against oxidative stress and immune escape. After briefly explaining its mechanism and its roles in cancer cell survival, the Authors outlined how macropinocytosis has been exploited for targeted cancer therapy using nanoparticles, viral vectors and exosomes, as well as for vehiculation of antibody- and peptide-drug conjugates.
The article is valuable in describing the potential applications of macropinocytosis in selective drug delivery, but a few aspects should be improved before publication. First, I would suggest to add a paragraph describing how macropinocytosis may be identified as the main mechanism of particle/conjugate internalisation and which molecules or processes can be used to either potentiate or inhibit it. Then, in some cited references macropinocytosis is not even mentioned, for example in lines 216-219: are all these albumin conjugates internalised by macropinocytosis? Is it the only mechanism? The same also applies to lines 269-276, 343-353.
Also, the Authors in lines 81-84, 128-133 state that macropinocytosis inhibition might be exploited as an anticancer therapy, so I would suggest to introduce a separate paragraph reporting current research on this topic, if available besides ref. 28, to make this review more complete.
Moreover, the manuscript must be thoroughly checked for mistakes, such as in line 51 (micropinocytic is a mistake?), 59 (micro..again), 70 (protein written with capital letter), 75 (add a comma after protein), 80 (space missing), 124 (leading instead of leads), 192 (enhanced instead of enhances), 197 (have instead of has), 228 (HDL, but in the following line LDL is mentioned).
Some acronyms have been used only once and so are useless, such as MP, FA, LDE, PLGA; some have not been explained (Tat-pDNA, LPP).
In line 336 the Authors state that Abraxane uses macropinocytosis for internalisation: a reference should be added.
The charge of the ions must be written superscript.
In figure and table captions “this is a figure”, “this is a table” should have been removed before submission.
Finally, the list of abbreviations should be reported in alphabetic order.
Reviewer 3 Report
Comments and Suggestions for Authors
This review aims to discuss the role of macropinocytosis (MP) in cancer and highlights how this mechanism can be considered both as a target for anticancer therapies or exploited for the delivery of anticancer therapeutics. Although the topic is of potential interest, this manuscript presents several critical issues both in form and content. In particular, many sentences are not pertinent, are scientifically inaccurate or express inexact concepts. Some significant examples:
- "The pharmacological properties of drug-monomer conjugation allow for faster serum clearance while maintaining cellular impermeability, thereby enhancing drug uptake through macropinocytosis";
- "Zhao et al. found that macropinocytosis inhibition compromises megakaryocyte (MK) differentiation, contributing to thrombocytopenia and ocular toxicity in cancer patients following ADC treatment [100,101]"; actually, this is not what references 100 and 101 demonstrate;
- The paragraph between lines 347 and 353 briefly describes the ADC sacituzumab govitecan as an example of macropinocytosis-mediated anti-cancer drug delivery system, without considering that ADCs primarily have targeted activity and act via receptor-mediated internalization, not macropinocytosis.
Other elements denote little care in drafting the manuscript. For example:
- The entire paragraph "3.1. Targeting macropinocytosis for cancer therapy" is missing, there is only the title (line 160);
- The numbering of the Tables is inaccurate: Tables start with 2 and 3, Table 1 is missing;
- Table 3 is never cited in the text;
- Many of the abbreviations listed on pages 10 and 11 are never found in the text (e.g. ABC, α-KG, CQ, CK2, etc.);
- The credited authors are six, but only four are mentioned in the "Author Contributions" section;
For these reasons, it is believed that the manuscript in its current form does not reach an adequate standard to be considered for publication in Biomolecules.
Round 2
Reviewer 1 Report
Comments and Suggestions for Authors
looks good now
Reviewer 2 Report
Comments and Suggestions for Authors
In the revised version of the manuscript the Authors have addressed all the remarks I had reported in my first review, so in my opinion the paper is suitable for publication in Biomolecules in the present form.
Reviewer 3 Report
Comments and Suggestions for Authors
The manuscript may be considered for publication in this revised form.